# Labeling Neural Representations with Inverse Recognition

**Kirill Bykov**[*]
UMI Lab
ATB Potsdam
Potsdam, Germany
kbykov@atb-potsdam.de

**Laura Kopf**
UMI Lab
ATB Potsdam
Potsdam, Germany
lkopf@atb-potsdam.de

**Shinichi Nakajima**
Machine Learning Group
TU Berlin
Berlin, Germany
nakajima@tu-berlin.de

**Marius Kloft**
Machine Learning Group
RPTU Kaiserslautern-Landau
Kaiserslautern, Germany
kloft@cs.uni-kl.de

**Marina M.-C. Höhne**
UMI Lab
ATB Potsdam
University of Potsdam, Germany
mhoehne@atb-potsdam.de

## Abstract

Deep Neural Networks (DNNs) demonstrate remarkable capabilities in learning complex hierarchical data representations, but the nature of these representations remains largely unknown. Existing global explainability methods, such as Network Dissection, face limitations such as reliance on segmentation masks, lack of statistical significance testing, and high computational demands. We propose Inverse Recognition (INVERT), a scalable approach for connecting learned representations with human-understandable concepts by leveraging their capacity to discriminate between these concepts. In contrast to prior work, INVERT is capable of handling diverse types of neurons, exhibits less computational complexity, and does not rely on the availability of segmentation masks. Moreover, INVERT provides an interpretable metric assessing the alignment between the representation and its corresponding explanation and delivering a measure of statistical significance. We demonstrate the applicability of INVERT in various scenarios, including the identification of representations affected by spurious correlations, and the interpretation of the hierarchical structure of decision-making within the models.

## 1 Introduction

Deep Neural Networks (DNNs) have demonstrated exceptional performance across a broad spectrum of domains due to their ability to learn complex, high-dimensional representations from vast volumes of data [1]. Nevertheless, despite these impressive accomplishments, our comprehension of the concepts encoded within these representations remains limited. The "black-box" nature of representations, combined with the known susceptibility of networks to learn spurious correlations [2, 3, 4], biases [5] and harmful stereotypes [6] poses significant risks for the application of DNN systems, particularly in safety-critical domains [7].

To tackle the problem of the inherent opacity of DNNs, the field of Explainable AI (XAI) has emerged [8, 9, 10]. The *global* explanation methods aim to explain the concepts and abstractions learned within the DNNs representations. This is often achieved by establishing associations between neurons and human-understandable concepts [11, 12, 13, 14], or by visualizing the stimuli responsible

---

[*]Corresponding author.

37th Conference on Neural Information Processing Systems (NeurIPS 2023).

for provoking high neural activation levels [15, 16, 17, 18]. Such methods demonstrated themselves to be capable of detecting the malicious behavior and identifying the specific neurons responsible [19, 20].

In this work, we introduce the *Inverse Recognition* (INVERT) [2] method for labeling neural representations within DNNs. Given a specific neuron, INVERT provides an explanation of the function of the neuron in the form of a composition of concepts, selected based on the ability of the neuron to detect data points within the compositional class. Unlike previous methods, the proposed approach does not rely on segmentation masks and only necessitates labeled data, is not constrained by the specific type of neurons, and demands fewer computational resources. Furthermore, INVERT offers a statistical significance test to confirm that the provided explanation is not merely a random occurrence. We evaluate the performance of the proposed approach across various datasets and models, and illustrate its practical use through multiple examples.

## 2 Related work

*Post-hoc* interpretability, a subfield within Explainable AI, focuses on explaining the decision-making strategies of Deep Neural Networks (DNNs) without interfering with the original training process [21, 22]. Within the realm of post-hoc methods, a fundamental categorization arises concerning the scope of explanations they provide. *Local* explanation methods aim to explain the decision-making process for individual data points, often presented in the form of attribution maps [23, 24, 25]. On the other hand, *global* explanation methods aim to explain the prediction strategy learned by the machine across the population and investigate the purpose of its individual components [26, 27].

Inspired by principles from neuroscience [28, 29, 30], global explainability directs attention towards the in-depth examination of individual model components and their functional purpose [31]. Often, global explainability is referred to as *mechanistic interpretability*, particularly in the context of Natural Language Processing (NLP) [32, 33, 34, 35]. Global approach to interpretability allows for the exploration of concepts learned by the model [36, 37, 38, 39] and explanation of *circuits* — computational subgraphs within the model that learn the transformation of various features [40, 41]. Various methods were proposed to interpret the learned features, including Activation-Maximisation (AM) methods [15]. These methods aim to explain what individual neurons or groups of neurons have learned by visualizing inputs that elicit strong activation responses. Such input signals can either be found in an existing dataset [16] or generated synthetically [42, 17, 18]. AM methods demonstrated their utility in detecting undesired concepts learned by the model [19, 43, 20]. However, these methods require substantial user input to identify the concepts embodied in the Activation-Maximization signals. Recent research has demonstrated that such explanations can be manipulated while maintaining the behavior of the original model [44, 45, 46].

Another group of global explainability methods aim to explain the abstraction learned by the neuron within the model, by associating it with the human-understandable concepts. The Network Dissection (NetDissect) method [11, 47] was developed to provide explanations by linking neurons to concepts, based on the overlap between the activation maps of neurons and concept segmentation masks, quantified using the Intersection over Union (IoU) metric. Addressing the limitation that neurons could only be explained with a single concept, the subsequent Compositional Explanations of Neurons (CompExp) method was introduced, enabling the labeling of neurons with compositional concepts [12]. Despite their utility, these methods generally have limitations, as they are primarily applicable to convolutional neurons and necessitate a dataset with segmentation masks, which significantly restricts their scalability (a more comprehensive discussion of these methods can be found in Appendix A.2). Other notable methods include CLIP-Dissect [13], MILAN [48], and FALCON [49]. However, these methods utilize an additional model to produce explanations, thereby introducing a new source of potential unexplainability stemming from the explainer model.

## 3 INVERT: Interpreting Neural Representations with Inverse Recognition

In the following, we introduce a method called *Inverse Recognition* (INVERT). This method aims to explain the abstractions learned by a neural representation by identifying what *compositional concept* representation is most effective at detecting in a binary classification scenario. Unlike the general

---

[2]The code can be accessed via the following link: `https://github.com/lapalap/invert`.

objective of Supervised Learning (SL) [50], which is to learn representations that can detect given concepts, the central idea behind INVERT is to learn a compositional concept that explains a given representation the best.

Let $\mathbb{D} \subset \mathbb{R}^m$, where $m \in \mathbb{N}$ is the number of dimensions of data, be the input (data) space. We use the term *neural representations* to refer to a sub-function of a network that represents the computational graph from the input of the model to the scalar output (activation) of a specific neuron, or any combination of neurons, that results in a scalar function.

**Definition 1** (Neural representation). *A neural representation $f \in \mathbb{F}$ is defined as a real-valued function $f : \mathbb{D} \to \mathbb{R}$, which maps the data domain $\mathbb{D}$ to the real numbers $\mathbb{R}$. Here, $\mathbb{F}$ represents the space of real-valued functions on $\mathbb{D}$.*

Frequently, in DNNs, particular neurons, like convolutional neurons, produce multidimensional outputs. Depending on the specific needs of the application, these multidimensional functions can be interpreted either as a set of individual scalar representations or the neuron's output can be aggregated to yield a single scalar output, e.g. with pooling operations, such as average- or max-pooling. Unless stated otherwise, we utilize average-pooling as the standard aggregation measure.

We define a *concept* as a mapping that represents the human process of attributing characteristics to data.

**Definition 2** (Concepts). *A concept $c \in \mathbb{C}$ is defined as a binary function: $c : \mathbb{D} \longrightarrow \{0, 1\}$, which maps the data domain $\mathbb{D}$ to the set of binary numbers. A value of $1$ indicates the presence of the concept in the input, and $0$ indicates its absence. Here, $\mathbb{C}$ corresponds to the space of all concepts, that could be defined on $\mathbb{D}$.*

In practice, given the dataset $\mathcal{D} \subset \mathbb{D}$, concepts are usually defined by labels, which reflect the judgments made by human experts. We define $C = \{c_1, ..., c_d\} \subset \mathbb{C}$ as a set of $d \in \mathbb{N}$ atomic concepts, that are induced by labels of the dataset (also referred to as *primitive concepts* or *primitives*). Within the context of this work, we permit concepts to be non-disjoint, signifying that each data point may have multiple concepts attributed to it. Additionally, we define a vector $\mathcal{C} = [c_1, \ldots, c_d] \in \mathbb{C}^d$.

A key step for explaining the abstractions learned by neural representations relies on the choice of the similarity measure between the concept and the representation. INVERT evaluates the relationship between representation and concepts by employing the non-parametric Area Under the Receiver Operating Characteristic (AUC) metric, measuring the representation's ability to distinguish between the presence and absence of a concept.

**Definition 3** (AUC similarity). *Let $f \in \mathbb{F}$ be a neural representation, dataset $\mathcal{D} \subset \mathbb{D}$ and concept $c \in \mathbb{C}$. We define a similarity measure $d : \mathbb{F} \times \mathbb{C} \longrightarrow [0, 1]$ as*

$$d(f, c) = \frac{\sum_{\{\, x \;|\; x \in \mathcal{D}, c(x) = 0 \,\}} \sum_{\{\, y \;|\; y \in \mathcal{D}, c(y) = 1 \,\}} \boldsymbol{I}\left[f(x) < f(y)\right]}{|\, \{\, x \;|\; x \in \mathcal{D}, c(x) = 0 \,\} \,| \cdot |\, \{\, y \;|\; y \in \mathcal{D}, c(y) = 1 \,\} \,|}, \tag{1}$$

*where $\boldsymbol{I}\left[f(x) < f(y)\right]$ is an indicator function that yields 1 if $f(x) < f(y)$ and 0 otherwise.*

AUC provides an interpretable measure to assess the ability of the representation to systematically output higher activations for the datapoints, where the concept is present. An AUC of $1$ denotes a perfect classifier, while an AUC of $0.5$ suggests that the classifier's performance is no better than random chance.

Given that various concepts have different numbers of data points associated with them, for concept $c \in \mathbb{C}$ we can compute *concept fraction*, corresponding to the ratio of data points that are positively labeled by the concept:

$$T(c) = \frac{|\, \{\, x \;|\; x \in \mathcal{D}, c(x) = 1 \,\} \,|}{|\, \{\, x \;|\; x \in \mathcal{D} \,\} \,|}. \tag{2}$$

## 3.1 Finding Optimal Compositional Explanations

Given a representation $f \in \mathbb{F}$, the INVERT's objective is to identify the concept, that maximizes the AUC similarity with the representation, or, in other words finding the concept that representation is detecting the best. Due to the ability of representations to detect shared features across various concepts explaining a representation with a single atomic concept from $C$ may not provide a comprehensive explanation. To surmount this challenge, we adopt the existing *compositional concepts*

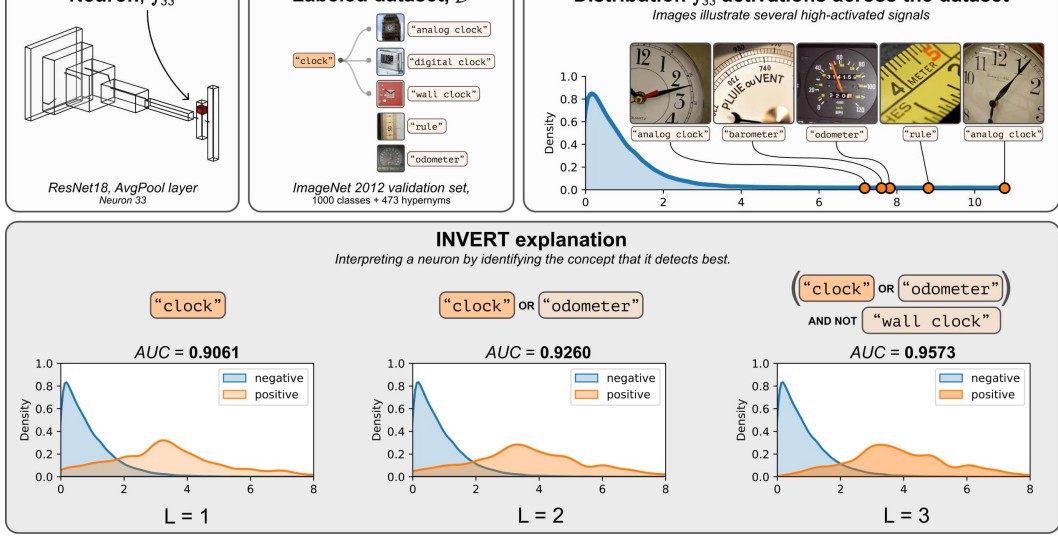

Figure 1: Demonstration of the INVERT method ($B = 1, \alpha = 0.35\%$) for the neuron $f_{33}$ from ResNet18, AvgPool layer (Neuron 33), using ImageNet 2012 validation dataset. The resulting explanations can be observed in the bottom part of the figure, where three steps of the iterative process are demonstrated from $L = 1$ to $L = 3$. It can be observed that INVERT explanations align with the neurons high-activating images, illustrated in the top right figure.

approach [12], and we augment the set of atomic concepts $C$ by introducing new generic concepts, as a logical combination of existing ones. These logical forms involve the composition of AND, OR, and NOT operators, and they are based on the atomic concepts from $C$.

**Definition 4** (Compositional concept). *Given a vector of atomic concepts $\mathcal{C}$, a compositional concept $\varphi$ is a higher-order interpretable function that maps $\mathcal{C}$ to a new, compositional concept:*

$$\varphi : \mathbb{C}^d \longrightarrow \mathbb{C}. \tag{3}$$

For example, let $C = \{c_1, c_2\}$ be a set of atomic concepts with corresponding vector $\mathcal{C}$. Let $c_1$ be a concept for "dog", and $c_2$ a concept for "llama". Then $\varphi(\mathcal{C}) = c_1$ OR $c_2 =$ "dog" OR "llama" is a compositional concept with the length $L = 2$. The $\varphi(\mathcal{C})$ is a concept in itself (i.e. $\varphi(\mathcal{C}) \in \mathbb{C}$) and corresponds to a concept that is positive for all images of dogs or llamas in the dataset.

Evaluating the performance of all conceivable logical forms across all of the $d$ concepts from $C$ is generally computationally infeasible. Consequently, the set of potential compositional concepts $\Phi_L$ is restricted to a form of predetermined length $L \in \mathbb{N}$, where $L$ is a parameter of the method. The objective of INVERT, in this context, can be reformulated as:

$$\varphi^* = \arg\max_{\varphi \in \Phi_L} d\left(f, \varphi(\mathcal{C})\right). \tag{4}$$

To determine the optimal compositional concept that maximizes AUC, we employ an approach similar to that used in [12], utilizing Beam-Search optimization. Parameters of the proposed method include predetermined length $L \in \mathbb{N}$, the beam size $B \in \mathbb{N}$. Additionally, during the search process explanations could be constrained to the condition $T(\varphi(\mathcal{C})) \in [\alpha, \beta]$, where $0 \leq \alpha < \beta \leq 0.5$. In Section 4.1, we further demonstrate that by imposing a such constraint on the concept fraction resulting explanations could be made more comprehensive. We refer to the standard approach when $\alpha = 0, \beta = 0.5$. In our experiments, unless otherwise specified, the parameter $\beta$ is set to 0.5. Additional details and a description of the algorithm can be found in Appendix A.3.

Figure 1 illustrates the INVERT pipeline for explaining the neuron from ResNet18 Average Pooling layer [51]. For this, we employed the validation set of ImageNet2012 [52] as the dataset $\mathcal{D}_I$ in the INVERT process. This subset contains 50,000 images from 1,000 distinct, non-overlapping

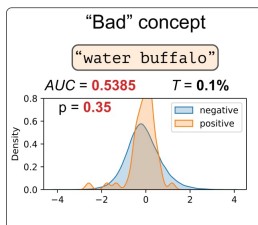
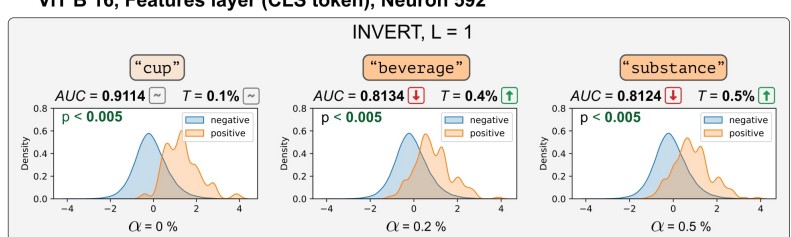

Figure 2: The figure illustrates the contrast between a *poor* explanation (on the left) and INVERT explanations with $L = 1$ and varying parameter $\alpha$, for neuron 592 in the ViT B 16 feature-extractor layer. The INVERT explanations were computed over the ImageNet 2012 validation set. The figure demonstrates that as the parameter $\alpha$ increases, the concept fraction $T$ also increases, indicating that more data points belong to the positive class. Furthermore, this figure showcases the proposed methods ability to evaluate the statistical significance of the result. The poor explanation fails the statistical significance test (double-sided alternative) with a p-value of 0.35, while all explanations provided by INVERT exhibit a $p < 0.005$.

classes, each represented by 50 images. Notably, since ImageNet classes are intrinsically linked to WordNet [53], we extracted an additional 473 hypernyms, or higher-level categories, and assigned labels for these overarching classes. In Figure 1 and subsequent figures, we use beige color to represent individual ImageNet classes and orange color to represent hypernyms. In the density plot graphs, the orange density illustrates the distribution of data point activations that belong to the explanation concept, while blue represents the distribution of activations of data points corresponding to the negation of the explanation.

## 3.2 Statistical significance

IoU-based explanations, such as those provided by the Network Dissection method [11], often report small positive IoU scores for the resulting explanations. This raises concerns about the potential randomness of the explanation. The AUC value is equivalent to the Wilcoxon-Mann-Whitney statistic [54] and can be interpreted as a measure based on pairwise comparisons between classifications of the two classes. Essentially, it estimates the probability that the classifier will rank a randomly chosen positive example higher than a negative example [55].

Given the concept $c \in \mathbb{C}$, this connection to the MannWhitney U test allows us to test if the distribution of the representations activations on the data points where concept $c$ is positive significantly differs from the distribution of activations on points where the concept is negative. We can then report the corresponding $p$-value (against a double- or one-sided alternative), which helps avoid misinterpretations due to randomness, thereby improving the reliability of the explanation process, as shown in Figure 2. In all subsequent figures, the explanations provided by INVERT achieve statistical significance (against double-sided alternative) with a standard significance level (0.05).

## 4 Analysis

In this section, we provide additional analysis of the proposed method, including the effect of constraining the concept fraction of explanations and comparison of the INVERT to the prior methods.

## 4.1 Simplicity-Precision tradeoff

The INVERT method is designed to identify the compositional concept that has the highest AUC similarity to a given representation. However, the standard approach neglects to account for the class imbalance between datapoints that belong and do not belong to a particular concept, often leading to *precise* but narrowly applicable explanations due to the small concept fraction. To mitigate this

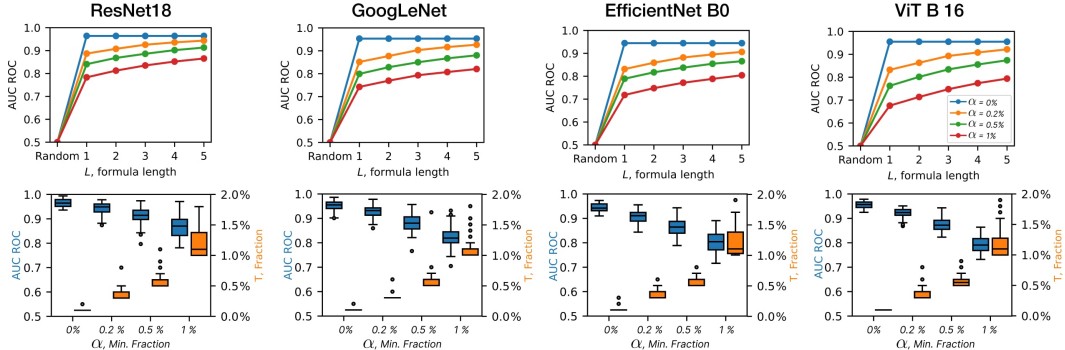

Figure 4: Impact of the parameter $\alpha$ and formula length $L$ on the resulting explanations. The first row of the figure shows the average AUC of optimal explanations for 50 randomly sampled neurons from the feature-extractor part of each one of the four ImageNet pre-trained models, conditioned by different values of parameter $\alpha$ in different colors. These graphs indicate that neurons generally tend to achieve the highest AUC for one individual class with $L = 1$ and $\alpha = 0$. The second row presents the distribution of AUC scores alongside the distribution of concept fractions $T$ for the INVERT explanations of length $L = 5$, for each model. Here, we can observe a clear trade-off between the precision of the explanation in terms of AUC measure and concept size $T$.

issue, we can modify the INVERT process to work exclusively with compositional concepts where the fraction equals or exceeds a specific threshold, represented as $\alpha$.

For this experiment, the INVERT method was utilized on the feature extractors of four different models trained on ImageNet. These models include ResNet18 [51], GoogLeNet [56], EfficientNet B0 [57], and ViT B 16 [58]. In this experiment, we examined 50 randomly chosen neurons from the feature-extractor layer of each model. We utilized the ImageNet 2012 validation dataset $\mathcal{D}_I$, which was outlined in the previous section, to generate INVERT explanations with $B = 3$ varying the explanation length $L$ between 1 and 5, and parameter $\alpha$, responsible for the constraining the concept fraction, $\alpha \in \{0, 0.002, 0.005, 0.01\}$.

The experiments results are depicted in Figure 4. For all models, we can see an effect that we call the *simplicity-precision tradeoff*: the explanations with the highest AUC typically involve just one individual class with a low concept fraction, achieved in an unrestricted mode with parameter $\alpha$ set to 0. By constraining the concept fraction $\alpha$ and increasing the explanation length $L$, we can improve AUC scores while still maintaining the desired concept fraction. Still, this indicates that more generalized, broader explanations come at the cost of a loss in precision in terms of the AUC measure. Figure 3 demonstrates how the change of parameter $\alpha$ affects the resulting explanation.

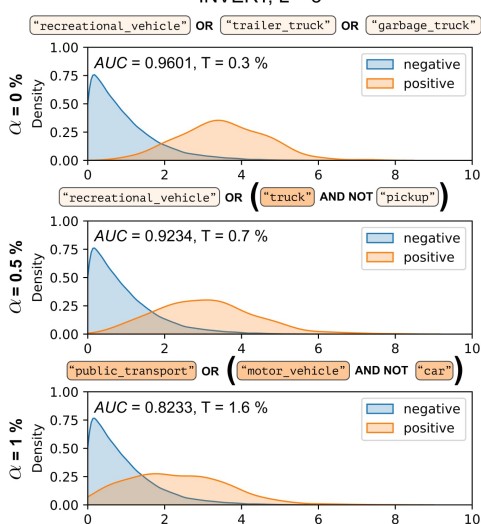

Figure 3: Three different INVERT explanations, computed by adjusting the parameter $\alpha$ for the Neuron 88 in ResNet18 AvgPool layer. Higher values of this parameter lead to broader explanations, albeit at the cost of precision, thus resulting in a lower AUC. The visualization of the WordNet taxonomy for the hypernyms is provided in the Appendix 3.

## 4.2 Evaluating the Accuracy of Explanations

While it is generally challenging to obtain ground-truth explanations for the latent representations in Deep Neural Networks (DNNs), in Supervised Learning, the concepts of the output neurons are

Table 1: A comparison of explanation accuracy between NetDissect and INVERT. The accuracy is computed by matching identified classes with the ground truth labels.

| Model | Dataset | NetDissect | INVERT |
|-------|---------|-----------|--------|
| MaskRCNN ResNet50 FPN | MS COCO | 95.06% | **98.77%** |
| FCN ResNet50 | MS COCO | **95.24%** | **95.24%** |
| ResNet18 | ImageNet | 19.2% | **73.2%** |
| GoogleNet | ImageNet | 19.7% | **82.2%** |
| DenseNet161 | ImageNet | 19.1% | **86.9%** |

defined by the specific task. In the subsequent experiment, we compared the performance of INVERT and Network Dissection in accurately explaining neurons when the ground truth is known.

For this experiment, we employed 5 different models: 2 segmentation models and 3 classification models. For image segmentation, we employed MaskRCNN ResNet50 FPN model [59], pre-trained on MS COCO dataset [60] and evaluated on a subset of 24,237 images of MS COCO train2017, containing 80 distinct classes, and FCN ResNet50 model [61], pre-trained on MS COCO, and evaluated on a subset of MS COCO val2017, limited to the 20 categories found in the Pascal VOC dataset [62]. For classification models we employed ImageNet pre-trained ResNet18 [51] DenseNet161 [63], and GoogleNet [64], with 1,000 output neurons, each neuron corresponding to the individual class in the ImageNet dataset.

The outputs from the segmentation models were converted into pixel-wise confidence scores. These scores were arranged in the format $[N_B, N_c, H, W]$, where $N_B$ represents the number of images in a batch, and $N_c$ signifies the number of classes. Each value indicates the likelihood of a specific pixel belonging to a particular class. To aggregate multidimensional activations, the INVERT method used a max-pool operation.

All the classification models that were used had 1,000 one-dimensional output neurons. The evaluation process for both explanation methods was carried out using a subset of 20,000 images from the ImageNet-2012 validation dataset. For the Network Dissection method, which necessitates segmentation masks, these masks were generated from the bounding boxes included in the dataset. Both Network Dissection and INVERT methods were implemented using standard parameters.

Table 1 presents the outcomes of the evaluation process. It is noteworthy that INVERT exhibits superior or equivalent performance to Network Dissection across all tasks. Importantly, INVERT can accurately identify concepts in image segmentation networks using only the labels of images, in comparison to the Network Dissection method that uses segmentation masks.

Figure 5: Comparing the computational cost of INVERT with Compositional Explanations of Neurons method (CompExp) in hours with varying formula lengths.

**Computational cost comparison**

Methods such as Network Dissection and Compositional Explanations (CompExp) of neurons have been observed to exhibit computational challenges mainly due to the operations on high-dimensional masks. While CompExp and INVERT share a beam-search optimization mechanism, the proposed approach allows for less computational resources since logical operations are performed on binary labels, instead of masks. Figure 5 showcases the running time of applying INVERT and Compositional Explanations for explaining 2048 neurons in layer 4 of the FCOS-ResNet50-FPN model [65] pre-trained on the MS COCO dataset [60] on a singe Tesla V100S-PCIE-32GB GPU.

The time comparison of varying formula lengths demonstrates the advantage of INVERT being more effective computationally, which leads to reduced running time and computational costs.

## 5 Applications

In this section, we outline some specific uses of INVERT, including auditing models for spurious correlations, explaining circuits within the models, and manually creating circuits with desired characteristics.

### 5.1 Finding Spurious Correlations by Integrating New Concepts

Due to the widespread use of Deep Neural Networks across various domains, it is crucial to investigate whether these models display spurious correlations, backdoors, or base their decisions on undesired concepts. Using the known spurious dependency of ImageNet-trained models on watermarks written in Chinese [19, 66, 67] we illustrate that INVERT provides a straightforward method to test existing hypotheses regarding the models dependency on specific features and allows for identification of the particular neurons accountable for undesirable behavior.

To illustrate this, we augmented the ImageNet dataset $\mathcal{D}_I$, with an additional dataset, $\mathcal{D}_T$, comprising 100 images. This new dataset contains 50 images for each of two distinct concepts: Chinese textual watermarks and Latin textual watermarks (see Appendix A.4). We created examples of these classes by randomly selecting images from the ImageNet dataset and overlaying them with randomly generated textual watermarks. Figure 6 depicts the change in the explanation process conducted on the original dataset and

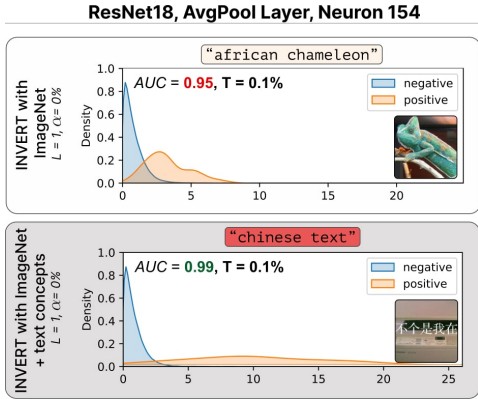

Figure 6: Difference of INVERT ($L = 1, \alpha = 0$) explanations of Neuron 154 in Average Pooling layer of ImageNet-trained ResNet18 model before (top) and after (bottom) integration of new concepts to the dataset.

its expanded version. Since the original dataset didn't include the concept of watermarked images, the label "African chameleon" was attributed to the representation. However, after augmenting the dataset with two new classes, the explanation shifted to the "Chinese text" concept, with the AUC measure increasing to 0.99. This demonstrates the capability of INVERT to pinpoint sources of spurious behavior within the latent representations of the neural network.

### 5.2 Explaining Circuits

INVERT could be employed for explaining circuits – computational subgraphs within the model, demonstrating the information flow within the model [41]. The analysis of circuits enables us to understand complex global decision-making strategies by examining how features transform from one layer to another. Furthermore, this approach can be employed for *glocal* explanations [68] – local explanation of a particular data point can be deconstructed into local explanations for individual neurons in the preceding layers, explained by INVERT.

To illustrate this, we computed INVERT explanations ($L = 3, \alpha = 0.002$) for all neurons in the average pooling layer of ResNet18. This was based on the augmented dataset from the preceding section. In ResNet18, the neurons in the Average Pooling layer have a linear connection to the output class logits. Figure 7 (left) illustrates the circuit of the three most significant neurons (based on the weight of linear connection) linked to the "carton" output logit. It could be observed that this class depends on Neuron 296, a "box" detector, and Neuron 154, which identifies the "Chinese text" concept. Furthermore, the right side of Figure 7 depicts the decomposition of local explanations: given an image of a carton box, we can dissect the GradCam [69] local explanation of a "carton" class-logit into the composition of local explanations from individual neurons. It is noticeable how Neuron 296 assigns relevance to the box, while Neuron 154 assigns relevance solely to the watermark present in the image. More illustrations of different circuits can be found in Appendix A.8.

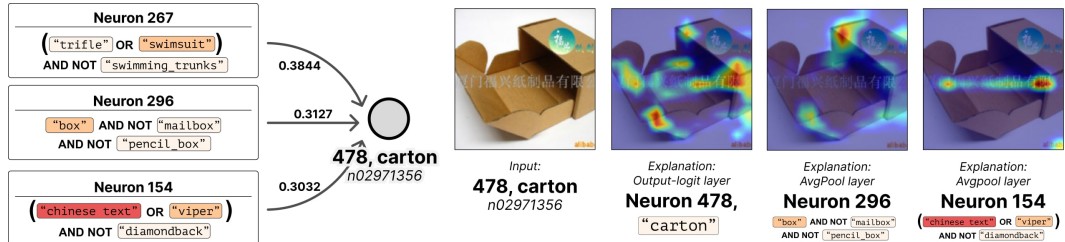

Figure 7: The figure illustrates the "carton" circuit within the ResNet18 model. The left part of the figure showcases the three most significant neurons (in terms of the weight of linear connection) and their corresponding INVERT explanation linked to the class logit "carton". The right part of the figure demonstrates how the local explanation from the class logit can be decomposed into individual explanations of individual neurons from the preceding layer.

## 5.3 Handcrafting Circuits

In this section, we demonstrate that its somewhat feasible to use the knowledge of what concepts are detected by neurons to combine them into manually designed circuits that can detect novel concepts. Just as compositional concepts are formed using logical operators, we employed fuzzy logic operators between neurons to construct meaningful handcrafted circuits with desired properties.

In contrast to conventional logic, fuzzy logic operators allow for the degree of membership to vary from 0 to 1 [70]. For this experiment, we employed the Gödel norm that demonstrated the best performance among other fuzzy logic operators (see Appendix A.5 for details). For the two functions $f, g : \mathbb{D} \longrightarrow [0, 1]$, the Gödel AND (T-norm) operator is defined as $\min(f, g)$ and the OR (T-conorm) is defined as $\max(f, g)$. Negation is performed by the $1 - f$ operation.

We utilized the ImageNet-trained ViT L 16 model [58], specifically 1024 representations from the feature-extractor layer. The output of each of these representations was mapped to the range $[0, 1]$ by first normalizing the output based on their respective mean and standard deviation across the ImageNet 2012 validation dataset, and then applying the Sigmoid transformation. In this experiment, for each of the 1473 ImageNet atomic concepts (which includes 1000 classes and 473 hypernyms), we identified a neuron from the feature-extractor layer that showed the highest AUC similarity. For instance, for the concept "boat", Neuron 61 exhibited the highest AUC similarity (denoted as $f_{\text{boat}}$), for the concept "house", Neuron 899 showed the highest AUC similarity (denoted as $f_{\text{house}}$), and for the concept "lakeside", Neuron 575 showed the highest AUC similarity (denoted as $f_{\text{lakeside}}$).

Further, we manually constructed six different compositional formulas using concepts from ImageNet that were designed to resemble different concepts from the Places365 [71] dataset. For example, for the "boathouse" class from Places365, we assumed that images from this class would likely include "boat", "house", and water, represented by the concept "lakeside". As such, we constructed a compositional formula "boat" AND "house" AND "lakeside" using concepts from the ImageNet dataset. Finally, using the neurons, that detect these concepts (e.g. $f_{\text{boat}}, f_{\text{house}}, f_{\text{lakeside}}$) we manually constructed the circuits using Gödel fuzzy logic operators. That is, for "boathouse" example, final circuit was formed as $g(x) = \min(f_{\text{boat}}(x), f_{\text{house}}(x), f_{\text{lakeside}}(x))$ using the Gödel AND operator. The performance of the resulting circuits was evaluated on the Places365 dataset in terms of AUC similarity with the concept. In essence, by labeling representations using the ImageNet dataset and manually building a circuit guided by intuition, we evaluated how this newly created function can perform in detecting a class in the binary classification task on a different dataset.

Figure 8 illustrates the "boathouse" example and three other handcrafted circuits derived from ViT representations (the other two circuits can be found in Appendix 16). We found that after performing this manipulation, the AUC performance in detecting the Places365 class improved compared to the performance of each individual neuron. This example shows that by understanding the abstractions behind previously opaque latent representations, we can potentially construct meaningful circuits and utilize the *symbolic* properties of latent representations. In Appendix A.6, we further demonstrate that when labels of the target dataset overlap or are similar to the dataset used for explanation, fine-tuning of the model can be achieved by simply employing representations with explanations matching the target labels.

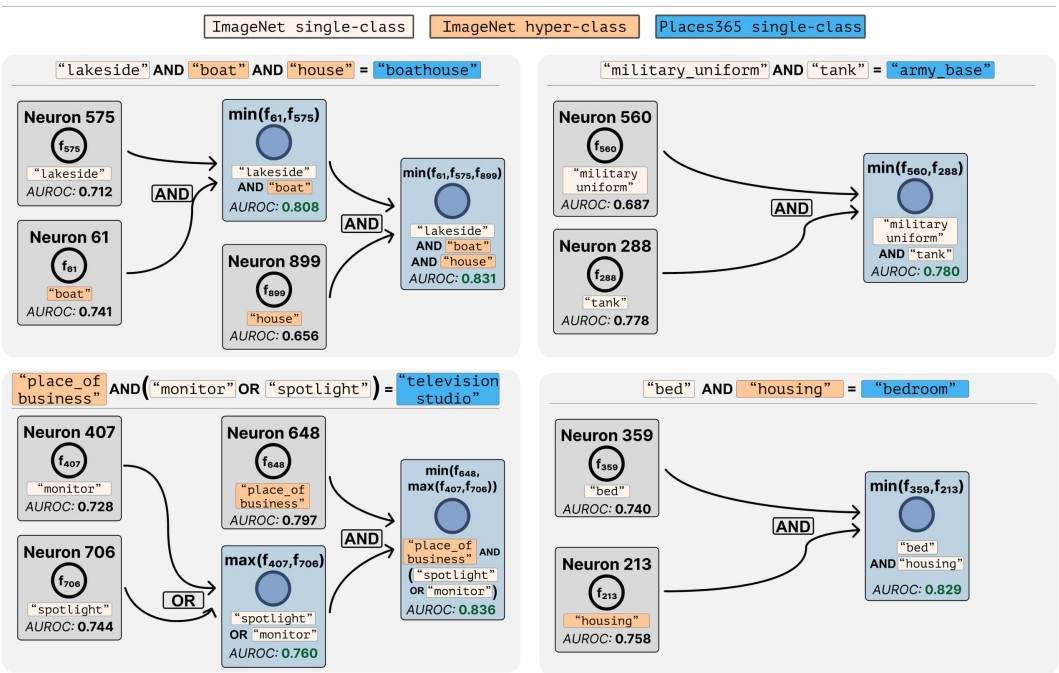

Figure 8: The figure presents four distinct handcrafted circuits, created from the latent representations from the ImageNet-trained ViT L 16 feature-extractor layer to detect classes from the Places365 dataset. For each neuron, or combination of neurons, we provide the Area Under the Receiver Operating Characteristic (AUROC) score for the Places365 concept in a binary classification task, distinguishing between the presence and absence of this concept.

## 6 Disscussion and Conclusion

In our work, we introduced the Inverse Recognition (INVERT) method, a novel approach for interpreting latent representations in Deep Neural Networks. INVERT efficiently links neurons with compositional concepts using an interpretable similarity metric and offers a statistical significance test to gauge the confidence of the resulting explanation. We demonstrated the wide-ranging utility of our method, including its capability for model auditing to identify spurious correlations, explaining circuits within models, and revealing *symbolic-like* properties in connectionist representations.

While INVERT mitigates the need for image segmentation masks, it still relies on a labeled dataset for explanations. In future research, we plan to address this dependency. Additionally, we will explore different similarity measures between neurons and explanations, and investigate new ways to compose human-understandable concepts.

The widespread use of Deep Neural Networks across various fields underscores the importance of developing reliable and transparent intelligent systems. We believe that INVERT will contribute to advancements in Explainable AI, promoting more understandable AI systems.

## Acknowledgements

This work was partly funded by the German Ministry for Education and Research (BMBF) through the project Explaining 4.0 (ref. 01IS200551). Shinichi Nakajima was supported by the German Ministry for Education and Research (BMBF) as BIFOLD - Berlin Institute for the Foundations of Learning and Data under the grant BIFOLD23B. Marius Kloft acknowledges support by the Carl-Zeiss Foundation, the DFG awards KL 2698/2-1, KL 2698/5-1, KL 2698/6-1, and KL 2698/7-1, and the BMBF awards 03|B0770E and 01|S21010C.

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

# A   Appendix

## A.1   Broader Impact

Our proposed INVERT method contributes to enhancing the transparency and safety of Deep Neural Networks. By providing human understandable and interpretable explanations for neurons in black-box models, our approach offers valuable insights into their internal operations, improving understanding. Moreover, our method is able to identify potentially spurious representations. An important advantage of our method is its notable reduction in computational cost compared to previous approaches. This reduction not only improves efficiency but also minimizes the harmful environmental impact associated with excessive GPU usage.

It is important to note that we cannot make definitive claims regarding specific groups of people benefiting from or being disadvantaged by our method. The general applicability and potential implications of our approach should be explored further and with caution.

## A.2   Prior work

Let's consider a function, $g : \mathbb{D} \to \mathbb{R}^{k \times k}$, that represents a convolutional neuron within a model that produces activation maps of dimensions $k \times k$, along with a concept $c \in \mathbb{C}$. Both Network Dissection [11] and Compositional Explanations of Neurons [12] methods make use of the Intersection over Union (IoU) similarity metric to measure the degree of correlation between a function and a concept. A prerequisite for these methodologies are segmentation masks of concepts, meaning for every concept $c \in \mathbb{C}$, there exists a corresponding function $M_c : \mathbb{D} \to \{0,1\}^{h \times w}$, which generates a binary mask for the specific concept, of the same size as the original input.

To evaluate the similarity between function $g$ and concept $c$, the multi-dimensional outputs from $g$ are subjected to thresholding based on neuron-specific percentiles (i.e., values above chosen percentiles are converted to 1 and the remaining to 0), and upscaled to match the dimensions of the original image. We can define the resulting function that produces binary masks of the same size as the input as $G : \mathbb{D} \to \{0,1\}^{h \times w}$. The final similarity (IoU) score between $g$ and $c$ can be computed as the Intersection over Union score between concept masks $M$ and function $G$ :

$$d_{\text{IoU}}(g,c) = \frac{\sum_{x \in \mathcal{D}} \mathbf{1}\left(M_c(x) \cap G(x)\right)}{\sum_{x \in \mathcal{D}} \mathbf{1}\left(M_c(x) \cup G(x)\right)}. \tag{5}$$

In section 4.2, the method of Compositional Explanations of neurons was applied using a 7x7 input map for each feature. Conversely, the INVERT approach uses a strategy that computes a scalar value by calculating the average of the input map.

## A.3   INVERT algorithm

Given a neural representation $f : \mathbb{D} \longrightarrow \mathbb{R}$, a dataset $\mathcal{D} \subset \mathbb{D}$, a set of atomic concepts $C \in \mathbb{C}$, and a vector $\mathcal{C} \in \mathbb{C}^d$ the INVERT approach seeks to identify a compositional concept $\varphi^*$, which is formed as a logical operation on the concepts, to optimize AUC similarity $d(f, \varphi^*(\mathcal{C}))$. For this purpose, we utilized an optimization process similar to that of the CompExpl methodology [12], employing Beam search to find the optimal compositional concept.

This method requires the configuration of certain parameters, namely the predetermined formula length $L \in \mathbb{N}$, the beam size $B \in \mathbb{N}$, and additionally, the parameters $\alpha, \beta$. Beam search intends to iteratively combine concepts, starting with the atomic concepts (primitives) from $C$. At every iteration of the process, the top $B$ best-performing compositional concepts are selected, and all feasible formulas are computed with primitives (i.e. atomic concepts). Subsequently, only the top $B$ best-performing concepts are selected, and the process continues until the formula reaches the predetermined length.

In detail, firstly, we define a set of primitives $\bar{\Phi}$ — a set of compositional concepts that correspond to the set of concepts $C$ and their negation. The set $\bar{\Phi}$ comprises $2k$ compositional concepts, with each concept corresponding to either the base concept or its negation. Next, all $2k$ concepts are evaluated in terms of AUC similarity with a given function, and the top $B$ best performing compositional concepts, that satisfy $\alpha \le T(\varphi(\mathcal{C})) \le \beta$ are selected, leading to the formation of the set $\Phi^*$ where

$|\Phi^*| = B$, referred to as a Beam. These are the top $B$ best-performing compositional concepts with a length of 1, satisfying the requisite condition on their positive fraction in the dataset. Subsequently, the following operations are iteratively performed until the predetermined formula length $L$ is met:

1. Each of the $B$ compositional concepts in the beam $\Phi^*$ is combined with all primitives (concepts from $\bar{\Phi}$) using either the AND or OR operation, thereby augmenting the formula length by 1, resulting in a total of $4Bk$ new formulas.

2. All newly generated formulas are evaluated based on their similarity to the representation, and the beam $\Phi^*$ is updated to include the top $B$ performing formulas, which satisfy the condition $\alpha \leq T(\varphi(\mathcal{C})) \leq \beta$.

Upon reaching the predetermined formula length $L$, the Beam-Search procedure concludes by identifying the compositional concept $\varphi^*$ with the highest observed AUC.

## A.4  Integrating Datasets from Different Sources

Let $\mathcal{D}_1$ and $\mathcal{D}_2$ be two separate datasets. Each of these datasets is linked to its unique set of concepts, represented as $C_1$ and $C_2$ respectively. By merging these datasets, we can form a consolidated dataset, symbolized as $\tilde{\mathcal{D}} = \mathcal{D}_1 \cup \mathcal{D}_2$. This unified dataset will encompass a combined set of concepts, denoted as $\tilde{C} = C_1 \cup C_2$.

The key requirement for this integration is the mutual definition: the concepts in $C_1$ should be defined within the dataset $\mathcal{D}_2$, and conversely, the concepts in $C_2$ should be defined within the dataset $\mathcal{D}_1$. While this does necessitate supplementary labeling, it becomes straightforward when it is evident that the concepts from both datasets do not overlap semantically. For instance, flower concepts from the Oxford Flowers102 [72] and faces from the CelebA [73] can be effortlessly combined. This is accomplished by designating the output of concepts within the non-native dataset as negative.

## A.5  Comparing Fuzzy Logic operators

Fuzzy logic operators [70] serve as essential instruments within the domain of fuzzy logic, a mathematical construct designed for modeling and handling data that is imprecise or vague. This contrasts with conventional logic where an element strictly either belongs to a set or not; fuzzy logic allows for the degree of membership to vary from 0 to 1, thereby allowing for partial membership.

In this experiment, our objective was to compare different fuzzy logic operators and examine their behavior concerning the proposed AUC metric. To fulfill this aim, we employed four distinct pre-trained deep learning image classification models: AlexNet [74], DenseNet161 [63], EfficientNet B4 [57], and ViT 16 L [58]. We focused on 1000 neural representations corresponding to the ImageNet classes in the output logit (pre-SoftMax) layer for each model, for which we recognized the 'ground-truth' concept — the corresponding ImageNet class. For fuzzy logic operators' testing, we mapped the output of each representation to the set $[0, 1]$ by normalizing each representation's output using their corresponding mean and standard deviation across the ImageNet dataset and applied a Sigmoid transformation. We tested four different Fuzzy logic operators, specifically Gödel, Product, ukasiewicz, and Yager with parameter $p = 2$, as illustrated in Table 3.

For performance evaluation, we generated random compositional concepts of a given length and computed the AUC similarity between fuzzy logic norms applied to functions corresponding to these concepts. For instance, given the random compositional concept $\varphi = c_i$ OR $c_j$, we derive *compositional representations* as per each of the four examined methods (e.g., the Gödel operator produces a function $h_G = \max(f_i, f_j)$). These compositional representations are then evaluated in terms of AUC similarity with the compositional concept — $d(h_G, \varphi)$.

We conducted the evaluation in two modes, that is, assessing the performance of the OR (T-conorm) operator and the performance of the AND (T-norm) operator. For each mode, we assembled 1000 random compositional concepts by sampling $L$ random concepts without replacement and calculated the AUC between compositional concepts and corresponding function. Note that for the second mode, AND (T-norm), random compositional concepts were assembled using the AND NOT operation, given the mutual exclusivity of ImageNet labels.

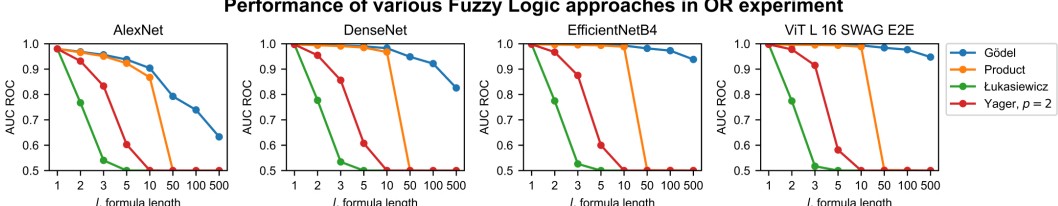

Figure 9: Average AUC similarity between random compositional OR concepts and corresponding compositional representations employing various Fuzzy logic operators (Higher is better) evaluated across four distinct models.

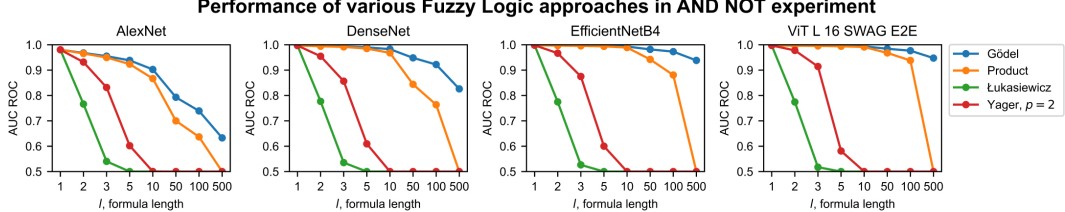

Figure 10: Average AUC similarity between random compositional AND NOT concepts and corresponding compositional representations employing various Fuzzy logic operators (Higher is better) evaluated across four distinct models.

Figures 9 and 10 depict the mean AUC similarity between random compositional concepts of varying lengths and the corresponding compositional representations, which were assembled using four distinct fuzzy logic operators. From these figures, it becomes evident that Gödel fuzzy logic operators demonstrate the most significant robustness to the length of the formula, consistently attaining superior AUC in contrast to other operators. Consequently, we can infer that Gödel's operator emerges as the optimal choice for implementing fuzzy logic operations on neural representations.

## A.6 Finetuning without training

In this section, we investigate whether it is feasible to perform model fine-tuning without having access to the target dataset, relying solely on the explanations of the latent representations and target class descriptions. In simple terms, the idea was to directly use the latent representation from ImageNet-trained models as a classificator for a class in another dataset that has a similar meaning to the explanation of the representation.

For this purpose, we utilized four different ImageNet deep learning image classification models, specifically AlexNet [74], DenseNet161 [63], EfficientNet B4 [57], and ViT 16 L [58], all of which were pre-trained on the ImageNet dataset. The *feature-extractor* layer that precedes the final output logit layer was used in all these models for our experiments. We computed the AUC similarity scores for all representations in each of the feature extractors in relation to ImageNet concepts.

Table 2: List of different fuzzy operators

|  | NOT$(a)$ | AND$(a,b)$ *(T-norm)* | OR$(a,b)$ *(T-conorm)* |
|---|---|---|---|
| *Gödel* | $1-a$ | $\min(a,b)$ | $\max(a,b)$ |
| *Product* |  | $a \cdot b$ | $a + b - a \cdot b$ |
| *Łukasiewicz* |  | $\max(a+b-1, 0)$ | $\min(a+b, 1)$ |
| *Yager, $p=2$* |  | $\max(1 - ((1-a)^2 + (1-b)^2)^{\frac{1}{2}}, 0)$ | $\min((a^2 + b^2)^{\frac{1}{2}}, 1)$ |

Table 3: A comparison of the accuracy achieved by the proposed finetuning method, which includes finetuning with a single representation (L=1), and multiple representations (L=2,5,10) combined with a fuzzy AND operator, against traditional and random finetuning baselines.

|  | AlexNet | DenseNet161 | EfficientNet B4 | ViT L 16 |
|---|---|---|---|---|
| Random | 2.21% | 2.21% | 2.21% | 2.21% |
| L =1 | 43.91% | 62.50% | 39.96% | 47.12% |
| L = 2 | 42.95% | 70.51% | 69.23% | 60.79% |
| L = 5 | 40.17% | 75.00% | 80.88% | 78.31% |
| L = 10 | 30.88% | 69.12% | 86.11% | 79.49% |
| Finetuned | 91.67% | 97.76% | 94.76% | 98.29% |

The target dataset employed for this study was the Caltech101 dataset [75], which comprises of 101 image classification categories. Specifically, we utilized a subset of this dataset that includes 46 classes, each of which has an exact or very similar equivalent in ImageNet classes.

We aimed to create a model for classifying Caltech classes by selecting suitable latent representations from the feature extractor layers of ImageNet models, and directly linking them to Caltech class logits. For each model, we chose a representation with the highest AUC similarity to the ImageNet concept closest to the Caltech concepts. This resulted in a subset of 46 neurons per model, each neuron having the highest AUC for an ImageNet concept similar to a Caltech concept. These neurons were normalized by the ImageNet validation dataset's mean and standard deviation, and a Sigmoid activation function was applied to constrain outputs between 0 and 1. Neuron selection was solely based on ImageNet explanations, with no Caltech101 data utilized.

We also hypothesized that individual signals from feature extractor layers could be further enhanced by executing a continuous AND operation with other neurons that share a high AUC towards the concept. Table 3 presents the results of this procedure in terms of the accuracy achieved on the target dataset. For this task, the random accuracy stands at approximately 2%, while the conventional fine-tuning approachwhich freezes the feature extractor layer and trains a linear classification layer atop the feature extractorsachieves an accuracy of up to 98.29% (last row of the table). Remarkably, by simply linking the representation with the highest AUC towards the ImageNet concept from the latent layer to the CalTech101 output class logit using our approach ($L = 1$), we were able to attain a substantial non-random accuracy, peaking at 69.50% in the case of DenseNet161. Furthermore, by selecting top $L$ neurons that have the highest AUC towards ImageNet concepts and employing Gödel AND operator between representations, we observed that this typically improved the results, with the only exception being the AlexNet model where this strategy slightly reduced the accuracy.

### A.7 Comparison between IoU and AUC metrics

In our supplementary experiments comparing different models, we further investigate the correlation between AUC and IoU. Table 4 demonstrates the performance of our method INVERT (AUC) in comparison to NetDissect and Compexp (IoU) performed on different models and layers including varying formula lengths (N). Our analysis employs ResNet18 and DenseNet161 PyTorch models trained on the Places365 dataset [71], accessible through the Compositional Explanations of Neurons implementation[3]. Following their approach we apply the methods on the ADE20k subset of the Broden dataset on formula lengths of 1 to 3. The IoU and AUC scores are summarized as the average and standard deviation across all neurons in each selected model layer. From these results, we can observe, that optimal explanations from AUC (INVERT) and IoU (NetDissect, CompExpl) based methods do not necessarily maximize each other objective functions.

The results in Table 5 reveal a correlation between IoU and AUC scores in non-zero IoU cases across multiple models and layers. The metrics differ in their applications and are not as strongly aligned. The correlation scores represent the average and standard deviation of the Pearson and Spearman correlation statistics. For each neuron and each available concept, correlations were calculated between the IoU and AUC scores. The "Normal" scenario corresponds to the standard case, whereas the "Log" case refers to when a logarithmic transformation was applied to the IoU values, with an

---

[3] https://github.com/jayelm/compexp/tree/master

Table 4: Comparison of IoU and AUC performed on different models and layers including varying formula lengths (N). All models are trained on the Places365 dataset and the explanations were constructed based on the ADE20k subset of the Broden dataset. The table presents the average and standard deviation scores IoU and AUC scores across all neurons in the selected model layer.

| Model - Layer | N = 1 | | | |
| --- | --- | --- | --- | --- |
| | INVERT | | NetDissect | |
| | IoU | AUC | IoU | AUC |
| ResNet18 - Layer 4 | 0.0062±0.0123 | 0.8959±0.0691 | 0.0581±0.0318 | 0.8367±0.1155 |
| ResNet18 - Layer 3 | 0.0007±0.0022 | 0.8834±0.0780 | 0.0121±0.0012 | 0.5549±0.1660 |
| DenseNet161 - Features | 0.0016±0.0066 | 0.8928±0.0733 | 0.0364±0.0279 | 0.7448±0.1547 |
| DenseNet161 - Dense Block 4 | 0.0007±0.0017 | 0.9014±0.0655 | 0.0150±0.0034 | 0.6877±0.1582 |
| | N = 2 | | | |
| | INVERT | | CompExp | |
| | IoU | AUC | IoU | AUC |
| ResNet18 - Layer 4 | 0.0021±0.0062 | 0.9972±0.0037 | 0.0756±0.0369 | 0.8310±0.1016 |
| ResNet18 - Layer 3 | 0.0023±0.0042 | 0.9955±0.0077 | 0.0185±0.0014 | 0.5726±0.1332 |
| DenseNet161 - Features | 0.0042±0.0124 | 0.9958±0.0056 | 0.0455±0.0313 | 0.7248±0.1424 |
| DenseNet161 - Dense Block 4 | 0.0029±0.0059 | 0.9961±0.0058 | 0.0222±0.0040 | 0.6930±0.1127 |
| | N = 3 | | | |
| | INVERT | | CompExp | |
| | IoU | AUC | IoU | AUC |
| ResNet18 - Layer 4 | 0.0026±0.0079 | 0.9977±0.0030 | 0.0849±0.0391 | 0.8184±0.0995 |
| ResNet18 - Layer 3 | 0.0021±0.0038 | 0.9966±0.0057 | 0.0235±0.0016 | 0.5714±0.1084 |
| DenseNet161 - Features | 0.0035±0.0104 | 0.9967±0.0046 | 0.0497±0.0330 | 0.7132±0.1356 |
| DenseNet161 - Dense Block 4 | 0.0026±0.0054 | 0.9969±0.0048 | 0.0361±0.0231 | 0.6846±0.1045 |

Table 5: Correlation between IoU and AUC based on the score for each class per neuron. The models were pre-trained using the Places365 dataset and their performance was assessed on the ADE20k subset of the Broden dataset. The table presents the average and standard deviation of the Pearson and Spearman correlation statistics.

| Model - Layer | Normal | | Log(+eps) | |
| --- | --- | --- | --- | --- |
| | Pearson | Spearman | Pearson | Spearman |
| ResNet18 - Layer 4 | 0.3429±0.0682 | 0.3623±0.0945 | 0.4116±0.0885 | 0.3623±0.0945 |
| ResNet18 - Layer 3 | 0.2377±0.0911 | 0.2738±0.1121 | 0.3009±0.1180 | 0.2738±0.1121 |
| DenseNet161 - Features | 0.2681±0.0869 | 0.2787±0.1041 | 0.3156±0.1050 | 0.2787±0.1041 |
| DenseNet161 - Dense Block 4 | 0.2143±0.1039 | 0.2691±0.1626 | 0.2878±0.1573 | 0.2691±0.1626 |

additional epsilon value of 1e-4. Correlations were computed exclusively for concepts that showed non-zero IoU scores. We can observe, that for non-zero IoU scores there exists a small positive correlation between IoU and AUC scores.

To further comprehend the correlation of these metrics we investigate the case where AUC and IoU perform differently. In Figure 11, we present a case where explanations yielding 0 IoU scores are better aligned with the explanation goal. We provide evidence of IoU-based explanations resulting in low neuron activation, while INVERT achieves notable activation even when IoU scores are 0.

Figure 12 (a) shows a qualitative example of AUC and IoU scores across all concepts of Neuron 269 from layer 4 of ResNet18 trained on the Places365 dataset [71]. Each data point corresponds to one concept among the 1105 ADE20k atomic concepts sourced from the Broden dataset [76, 11]. This example illustrates the dependence between AUC and IoU, high IoU scores are correlated with high AUC scores. In Figure 12 (b) we showcase the top 4 most activating images for Neuron 269 from the ADE20k dataset to align them with the highest scoring concepts in Figure 12 (a). Comparing the set of images with the concepts exhibiting the highest AUC scores (e.g., "throne room", "apse-indoor", "fur"), we observe a strong visual alignment. However, when examining the concepts with high IoU scores (e.g., "nursery", "cradle", "attic"), we find a relatively low degree of visual similarity. Those results demonstrate the limitations of the IoU measure for evaluating explanations.

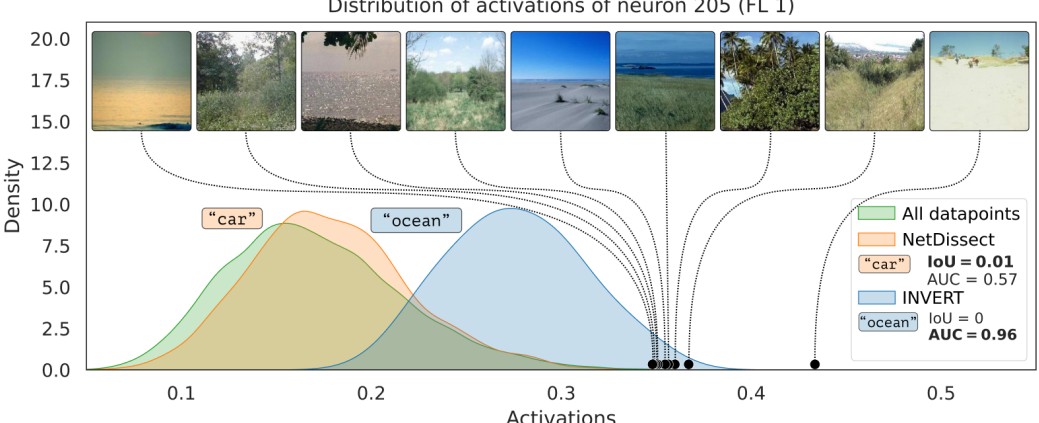

Figure 11: Comparison between INVERT and NetDissect. The figure displays three distributions of activations: one for all datapoints in green, one for datapoints corresponding to the IoU-based explanation in orange, and one for the AUC-based explanation in blue. These distributions pertain to the average activation across activation maps of Neuron 205 in ResNet18, layer 3, pre-trained on the Places365 dataset. The activations were collected across the ADE20k subset of the Broden dataset. The class labeled as "car" resulted from IoU optimization, while the class labeled as "ocean" resulted from AUC optimization. Notably, even though the "ocean" class has an IoU score of 0, it comprises some of the most activating images for the neuron, as evidenced by the top 9 most activated images.

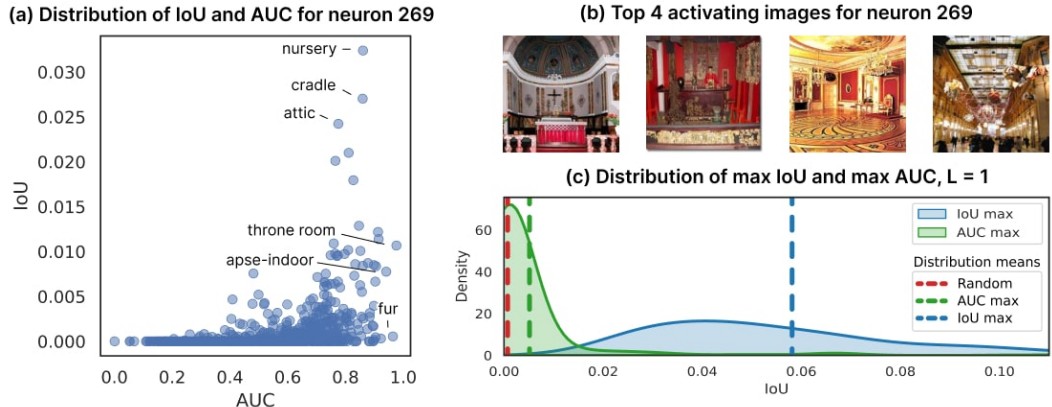

Figure 12: In (a) we compare the distribution of AUC and IoU across all concepts of the ADE20k atomic concepts from the Broden dataset for Neuron 269 from layer 4 of ResNet18 trained on the Places365 dataset, where (b) shows the top 4 activating images of the ADE20k dataset. (c) shows the distribution of maximized IoU, maximized AUC, and random IoU scores for layer 4 of ResNet18 trained on the Places365 dataset with a formula length of 1.

Table 6: Metric comparison for true label and random explanation evaluation. FasterRCNN ResNet50 FPN (pre-trained on MS COCO for object detection). UPerNet BEiT-B (pre-trained on ADE20k for semantic segmentation).

| Metric | FasterRCNN ResNet50 FPN | | UPerNet BEiT-B | |
|---|---|---|---|---|
| | True | Random | True | Random |
| IoU | 0.8355±0.0466 | 0.0077±0.0046 | 0.8553±0.0913 | 0.0007±0.0007 |
| AUC | 0.9556±0.0371 | 0.5005±0.0253 | 0.8738±0.0929 | 0.5001±0.0164 |

Furthermore, we conducted a quantitative evaluation shown in Figure 12 (c), specifically focusing on layer 4 of the ResNet18 trained on the Places365 dataset. We compared the distribution of IoU scores of explanations obtained by maximizing IoU and AUC respectively. Additionally, we examined the mean values of these distributions, which included random IoU scores as baseline reference. Our findings reveal that maximizing IoU leads to a relatively sparse distribution of IoU scores while maximizing AUC results in a more densely concentrated accumulation of predominantly low IoU scores. As anticipated, the performance of random IoU scores was notably poor. We can observe that maximizing AUC also indirectly maximizes IoU.

Table 6 serves as a sanity check implementing metric comparison for best explanation and random explanation evaluation. The FasterRCNN ResNet50FPN model was pretrained on the MS COCO dataset for object detection, while the UPerNet BEiT-B model was pretrained on ADE20k for semantic segmentation. The former model's evaluation was conducted on a subset of MS COCO containing 20,000 images, while the latter was assessed on the ADE20k subset of the Broden dataset. The output layers of both models were utilized to access the "ground truth" label for each neuron. In the table, the "True" column represents the IoU/AUC scores of the explanation that align with the ground-truth neuron label. On the other hand, the "Random" column corresponds to the scores of randomly chosen explanation-concept pairs that differ from the "ground truth".

## A.8  Figures

**WordNet taxonomy graph**

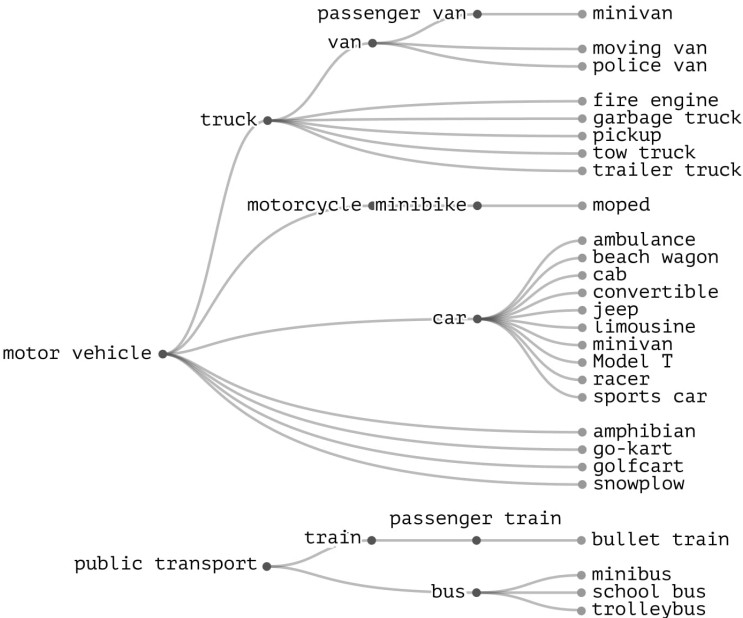

Figure 13: The figure displays the WordNet taxonomy, which was used to gather the hierarchical structure of the labels for the Figure 3.

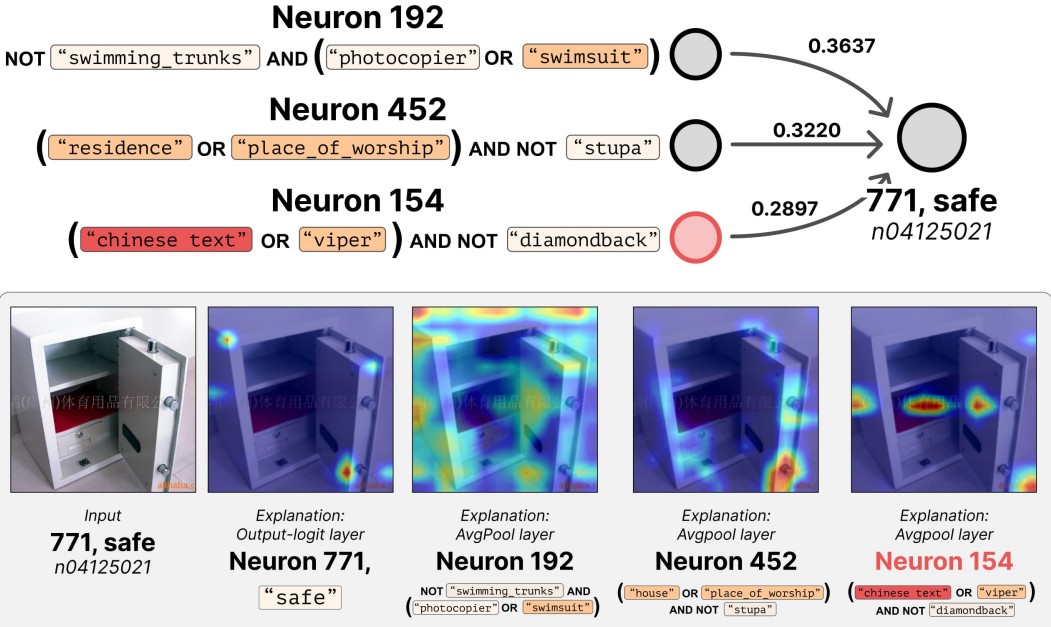

Figure 14: The figure illustrates the "safe" circuit within the ResNet18 model. The top part of the figure showcases the three most significant neurons (in terms of the weight of linear connection) and their corresponding INVERT explanation linked to the class logit "safe". The bottom part of the figure demonstrates how the local explanation from the class logit can be decomposed into individual explanations of individual neurons from the preceding layer. This allows for a more detailed understanding of how each neuron contributes to the final classification.

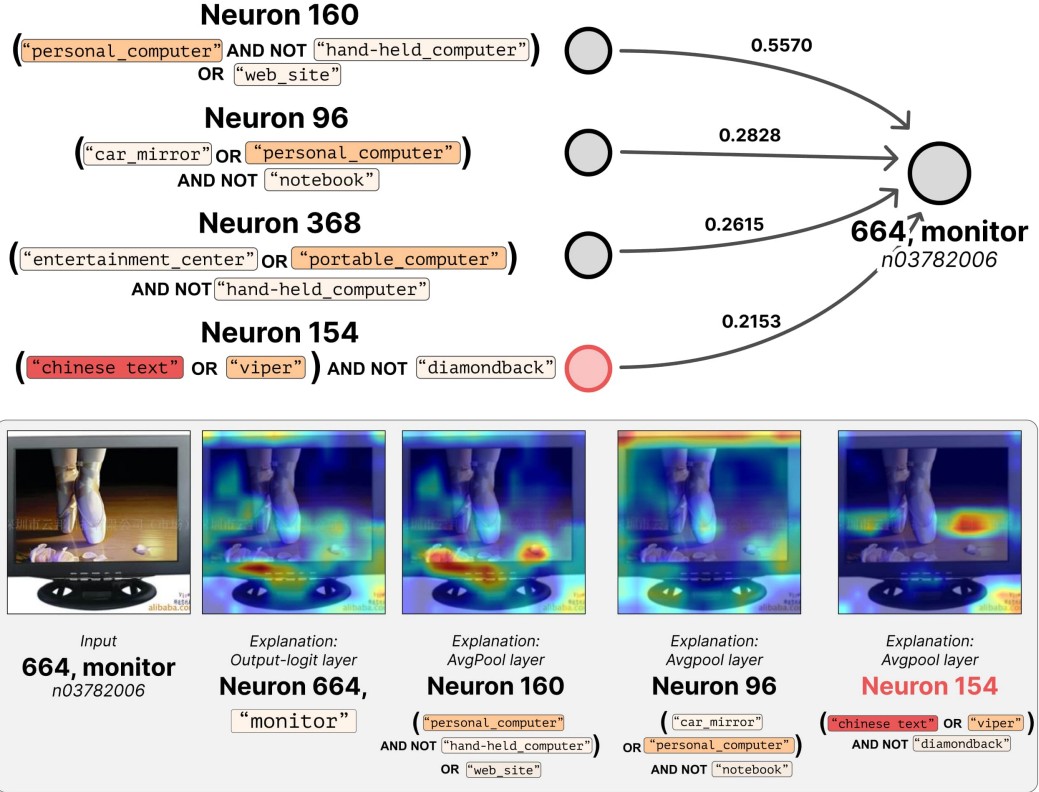

Figure 15: The figure illustrates the "monitor" circuit within the ResNet18 model. The top part of the figure showcases the four most significant neurons (in terms of the weight of linear connection) and their corresponding INVERT explanation linked to the class logit "monitor". The bottom part of the figure demonstrates how the local explanation from the class logit can be decomposed into individual explanations of individual neurons from the preceding layer.

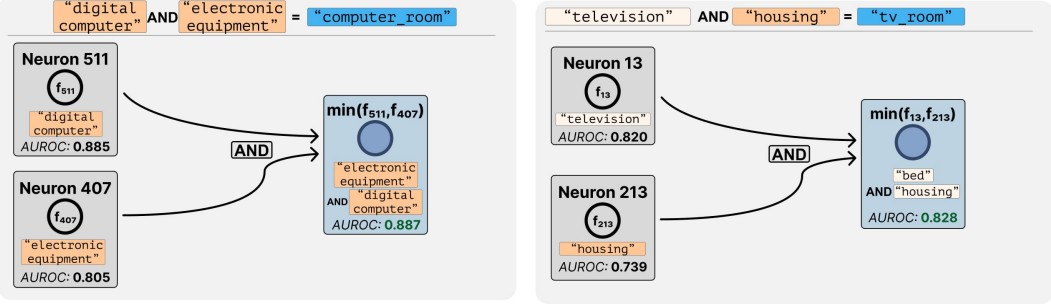

Figure 16: The figure presents two distinct handcrafted circuits. For each neuron, or combination of neurons, we report the Area Under the Receiver Operating Characteristic (AUROC) score. This score represents the AUC classification performance towards classifying specific concepts from the Places365 dataset.

