# OpenReview forum: "Labeling Neural Representations with Inverse Recognition"
_NeurIPS.cc/2023/Conference — NeurIPS 2023 poster_

### Official Review · Reviewer_LJJZ · 2023-07-03

**Soundness:** 2 fair
**Presentation:** 3 good
**Contribution:** 2 fair
**Rating:** 5
**Confidence:** 3

**Summary:**

The paper proposes a new explainability method, named INVERT, that matches the learned representations with human concepts. Specifically, it tries to provide explanations to individual neurons by matching human concepts that the neuron predominantly detects. Compared to previous approaches, INVERT is computationally more efficient, does not depend on the annotation of segmentation masks, and is generalizable to different types of neurons. The paper shows INVERTs applicability in identifying neurons affected by spurious correlations and fine-tuning representations without explicitly training them.

**Strengths:**

The method is generalizable in terms of architecture, ie not restricted to convolutional neurons as previous ones. The method is more efficient by requiring less running time and computational resources. The method does not require additional annotations, such as segmentation masks. The method is simple and easy to follow. From Section 4.2, INVERT seems to provide outputs more visually similar to human concepts.

**Weaknesses:**

My main concern about this work is the lack of evidence that this method generalizes to other configurations.
For instance, in the related work, one of the limitations that was described from previous work is the fact that they work only on convolutional neurons. However, most of the experiments are limited to ResNet-18 (except Section 5.2 which a ViT was used), which is mainly based convolutional layers. Since this is one of the important contributions of the method, it would be important to see how the results generalize to other types of neurons. Another suggestion is to expand the results shown in Figures 4 and 6 (for other neurons and architectures) and place them in the appendix.

Minor writing feedback:
1. Line 35, in “(1)” should add the word “Figure”, eg “(Figure 1)”.
2. Missing parenthesis on line 142 to close the min equation
3. Figure 2 (in page 5) is never cited in the paper
4. Would encourage the authors to refer to specific sections of the appendix when referring to it in the main text


**Questions:**

* Does the conclusions of Figure 4(c), that maximizing IoU leads to a relatively sparse distribution of IoU scores while maximizing AUC results in a more densely concentrated accumulation of low IoU scores, generalizes to other lengths as well (eg L=2 or L=3)? For lack of space, one suggestions would be add such results to appendix.
* In line 218, the conclusion of “high IoU scores are correlated with high AUC scores” from the Figure 4(a) is not clear to me since several points that have high AUC (x axis) also have low IoU (y axis). Could the authors use some form of correlation measure to produce a single score to better verify this statement?
* Section 5.2 focuses on “creating” a model trained on ImageNet to classify Caltech images without any training procedure. However, this seems limited to the setting where classes have some form of overlap (which is the case in this example, as also mentioned in the text), or when you can overlap them by combining different concepts (described at the last paragraph of this section). Instead of relying on this assumption, could INVERT be used to somehow demonstrate how well a set of representations will be transferred to another dataset without actually training the model (ie as an analysis tool, not as a model construction itself as it was done)? If yes, then this could be a powerful application of the method, since it could be used to analyze whether the costs of fine-tuning a model will be paid off or not. This could be shown by comparing two models, one that fine tunes well and another that doesn’t, and how this method can be used to predict that beforehand.

**Limitations:**

A section or a couple of sentences in the conclusion could be added to broadly discuss some of the limitations of the proposed method.

---

> ### Author Rebuttal · Authors · 2023-08-10
>
> We thank the reviewer LJJZ for the time spend reviewing our work and we are thankful for the detailed comments. In following, we will answer to the described shortcomings of our work and answer the raised questions.
>
> *The Lack of evidence that this method generalizes to other configurations.*
>
> **Answer:**
>  We thank the reviewer for raising the following a concern. Indeed, previous methodologies, such as Network Dissection and Compositional Explanation of Neurons, were designed to explain convolutional neurons through the examination of intersections between neuron activation maps and object masks. INVERT overcomes such limitations by analysing and explaining scalar functions. It is important to emphasize that our ResNet18 experiments, as detailed in the original paper, were conducted on the AveragePooling layer—a configuration that produces scalar activations (512 neurons x 1 activation). This already demonstrates a difference from prior IoU-based techniques designed for convolutional neurons, which generate high-dimensional activation maps.
> It's worth noting that INVERT exhibits versatility, extending its applicability to transformer-based architectures, including Visual transformer models (ViT). This was aptly demonstrated in Section 5.2 of our original paper. Our efforts to bolster this aspect continue with the inclusion of additional qualitative examples in the updated version. Many of these new visualizations can be found in the PDF file accompanying this global rebuttal. Additionally, the updated version features a showcase of "handcrafted" circuits founded on ViT representations, further solidifying our method's compatibility and efficacy.
>
> *Question: Does the conclusions of Figure 4(c), that maximizing IoU leads to a relatively sparse distribution of IoU scores while maximizing AUC results in a more densely concentrated accumulation of low IoU scores, generalizes to other lengths as well (eg L=2 or L=3)? For lack of space, one suggestions would be add such results to appendix.*
>
> **Answer**
> We have indeed performed the quantitative comparison between IoU-based and AUC-based explanations, focusing on the same neurons but with varying formula lengths. Although these specific results are not included in the attached PDF, they have been incorporated into the appendix of our paper. While there does exist a general correlation between IoU and AUC, as highlighted in Table 1 of the PDF, our investigations revealed that often optimal IoU explanations did not align with high AUC scores, and vice versa.
> Our global rebuttal, with particular emphasis on Figure 1, elucidates a compelling case where an AUC-based explanation yields 0 IoU, while the best IoU-based explanation demonstrates a low AUC score. We argue that due to the interpretability inherent in the AUC metric, the reduced computational complexity, the capacity to ascertain random explanations through statistical testing, and the broader applicability, INVERT emerges as a robust and preferable alternative to IoU-based methods.
>
> *Question:In line 218, the conclusion of “high IoU scores are correlated with high AUC scores” from the Figure 4(a) is not clear to me since several points that have high AUC (x axis) also have low IoU (y axis). Could the authors use some form of correlation measure to produce a single score to better verify this statement?*
>
> **Answer:**
>  We addressed this concern within our global rebuttal. To elaborate, we have conducted a quantitative evaluation that examines the correlations between IoU and AUC scores. As a result, we have relaxed our assertion in the updated version of our paper. Our analysis demosntrates a positive correlation between IoU and AUC scores. However, it is important to highlight that in instances where both IoU and AUC scores are at their highest, often metrics disagree, as illustrated in Figure 1 of the attached PDF, accompanying our global rebuttal.
>
>
> *Question: Empolying INVERT to demonstrate how well a set of representations will be transferred to another dataset.*
>
> **Answer:**
>  We find such application  very interesting, but we believe that it lies outside of the scope of the proposed approach.  INVERT allows to connect neurons with human-understandable concepts – however, given, for example, explanation “farm” based on ImageNet data, it is hard to quantify how good such neuron would detect “farm” classes from another dataset, without the access or evaluation on the “farm” images from another dataset. The assessment of the scope of global explanations, including the assessment of the ability of neuron to generalise to other datasets based on its explanation might be a promising avenue for the future ressearh.
>
> Minor mistakes
> We agree with these mistakes and we fixed them in the updated version of our paper.
>
> We extend our sincere gratitude to the reviewer for their thoughtful and comprehensive feedback. The insights you have provided have proven invaluable in refining our work. We firmly believe that the combined impact of our global rebuttal, fortified by new quantitative and qualitative findings, will positively influence the re-evaluation of our paper.

---

> > ### Comment · Reviewer_LJJZ · 2023-08-19
> >
> > Thanks authors for the detailed rebuttal. While I acknowledge the potential of such method, I will keep my original rating since I think there is room for improvement.

---

### Official Review · Reviewer_bbMD · 2023-07-06

**Soundness:** 3 good
**Presentation:** 3 good
**Contribution:** 3 good
**Rating:** 5
**Confidence:** 4

**Summary:**

This paper introduces INVERSE, a scalable approach called Inverse Recognition, that links learned representations to human-interpretable concepts by leveraging the ability to differentiate between concepts The applicability of INVERSE is demonstrated in diverse scenarios, including identifying representations influenced by spurious correlations and interpreting the hierarchical decision-making structure within the models.

**Strengths:**

1. The paper introduces INVERT, a novel approach called Inverse Recognition, which enables the labeling of neural representations with human-interpretable concepts in a scalable and informative manner.
2. The authors make significant contributions by using INVERT to gain insights into the hierarchical decision-making structure within models, enhancing our understanding of their inner workings. They also propose an interpretable metric to assess the alignment between representations and explanations, providing a means to evaluate explanation quality.
3. In addition to the aforementioned contributions, the authors demonstrate the practical applications of INVERT, further highlighting its usefulness and credibility.
4. The paper exhibits strong writing quality with a well-structured and logical flow. The equations are clearly presented, and the figures and tables are easily comprehensible, effectively conveying the authors' ideas.



**Weaknesses:**

1. While the paper showcases the practicality of INVERT in diverse scenarios, the evaluation is confined to a limited number of examples. Conducting a broader evaluation encompassing a wider range of models and datasets would be advantageous.
2. The paper offers a broad overview of the INVERT methodology, but certain aspects lack clarity. Providing a more detailed explanation of the methodology would enhance readers' understanding of its workings. Specifically, Section C in the Appendix could be better placed within Section 3 for improved organization and coherence.
3. One aspect that should be noted is that in INVERT, the concepts to be linked with the learned representations still need to be pre-defined or manually selected.

**Questions:**

1. What is the motivation ot intuiation using the logocal forms, such as AND, OR, and NOT?
2. Do you have some ideas or suggestions dealing with the tradeoff between simplicity and precision?

---

> ### Author Rebuttal · Authors · 2023-08-10
>
> We extend our gratitude to Reviewer bbMD for their insightful and constructive feedback. We greatly appreciate the time and effort invested in reviewing our work. We are heartened by the positive acknowledgment of the significance of our contribution, as well as the recognition of the quality of our paper's content and visual aids.
>
> Herein, we address the noted weakness and questions raised by the reviewer:
>
> *While the paper showcases the practicality of INVERT in diverse scenarios, the evaluation is confined to a limited number of examples. Conducting a broader evaluation encompassing a wider range of models and datasets would be advantageous.*
>
> **Answer**
> We acknowledge the reviewer's concern and, in response, we have taken substantial steps to address it. The revised version of our manuscript includes an expansion of our quantitative experiments. We have investigated the correlation between AUC and IoU measures for explanations across different model and layers, an analysis that contributes to a more comprehensive understanding of our approach. Moreover, we have investigated the behavior of random explanations, shedding light on an aspect critical for establishing the efficacy of our method. These updates are described in detail in a global rebuttal.
>
> *The paper offers a broad overview of the INVERT methodology, but certain aspects lack clarity. Providing a more detailed explanation of the methodology would enhance readers' understanding of its workings. Specifically, Section C in the Appendix could be better placed within Section 3 for improved organization and coherence.*
>
> **Answer**
> We thank the reviewer for raising this point. In the revised manuscript, we have shortened several paragraphs and allocated space for a more thorough discussion of the algorithm within Section 3, which greatly improve theclarity and coherence in the presentation of our methodology.
>
> *One aspect that should be noted is that in INVERT, the concepts to be linked with the learned representations still need to be pre-defined or manually selected.*
>
> **Answer**
> The issue of data-dependency is a shared characteristic among methods aimed at explaining the concepts learned by neural representations, including Network Dissection and Compositional Explanation of Neurons. While INVERT does indeed necessitate predefined concepts, it distinguishes itself by mitigating the reliance on masked data, relying instead on image labels. This shift enhances the feasibility of accessing a more extensive spectrum of concepts, considering that labeled data is more easily accessible than masked image data. Moreover, this approach accelerates computational processes. Section 5.1 of our paper delves into this matter, illustrating how the set of concepts can be broadened by merging datasets from diverse sources, effectively encompassing a wider array of concepts. We consider the research towards data-free explanation method to be an avenue for the future work.
>
> *Question: What is the motivation ot intuiation using the logocal forms, such as AND, OR, and NOT?*
>
> **Answer**
> Main inspiration for the logical forms approach comes from the paper “Compositional Explanation of Neurons”, where such approach was introduced. Such method allows to enrich the basic, atomic concepts with interpretale combinations, allowing for the more versatile explanations. Although alternate (from logical) forms of function $\varphi$ (as defined in Definition 4) are feasible, we opted for logical forms due to their established nature and interpretability. Our framework, however, is versatile enough to accommodate other $\varphi$ forms, which we plan to explore in future work.
>
> *Question: Do you have some ideas or suggestions dealing with the tradeoff between simplicity and precision?*
>
> **Answer**
> Response: The trade-off between simplicity and precision is a captivating challenge in the realm of explainable AI. Within the context of INVERT, we believe enhancing precision involves incorporating more detailed and precise concepts into the dataset. By introducing more concepts, the general behaviour of the neurons could be explained more precise, while still being comprehensible for a human. This, for example, could be achieved by aggregating diverse concepts from various sources, as outlined in Section 5.1.
>
> Once again, we express our gratitude for your insightful feedback, which significantly improved  the depth and impact of our work. We hope that the improvement of our work will contribute to a more favorable assessment of our work.

---

### Official Review · Reviewer_UsfE · 2023-07-07

**Soundness:** 2 fair
**Presentation:** 1 poor
**Contribution:** 2 fair
**Rating:** 3
**Confidence:** 5

**Summary:**

The paper introduces a method to understand what semantic concepts different neurons of a deep network are looking at. They do so by comparing the similarity between different concepts (taken from a pre-defined concept bank) and neuron representations using the AUROC metric, and assign the concept with the highest AUROC (to the neuron under study). The authors further illustrate how their uncovered concepts can be used to handcraft circuits for recognizing different classes from another dataset and detect spurious correlations.

**Strengths:**

I greatly appreciate the AUROC metric the authors presented. It extends current approaches to neuron interpretation based on segmentation maps and IOU metrics which are necessarily limited to convolution filters.

**Weaknesses:**

Unfortunately, I think the paper is poorly written with key details missing, prior work not appropriately mentioned and experimental validation of method lacking. Due to this I recommend rejection of this manuscript in its current form. Below I describe the weaknesses in detail in the questions section.

**Questions:**

1. **Incorrect Prior Work/Missing References:** While describing MILAN in line 73 "While alleviating the limitation posed by the necessity of having a labeled dataset". This statement isn't correct since MILAN does require a labelled dataset to train their captioning model (the MILANAnnotations dataset). I believe the authors meant to say human annotated segmentation masks are not required? The authors should also cite CLIP-Dissect which was published ICLR 2023, which did away with the need for labelled datasets by appealing to the zero-shot abilities of CLIP.

2. **missing details in figures:** The captions are unclear and sometimes not referenced in the main text so their interpretation is not clear. For instance, what is the density being plotted in figure 2a? Is it the density of neuron activations for images labelled positive for the concept and images labelled as negative for the concept?

3. **More rigorous comparison to prior work*** From figure 4a, it is not clear higher AUC is correlated with higher IOU, if anything it seems at IoU 0, AUC can be anything between 0 - 1. I would suggest the authors do a more rigorous study to substantiate their claim perhaps by computing a correlation coefficient. Moreover, figure 4b, is confusing it plots the distribution of iou for max iou? I understand the authors are trying to show that the explanation that maximizes the ioU is not necessarily concentrated and can be more spread out vs the distribution for max AUC is concentrated. However, the motivation for this experiment is not clear, nor is it clear from the discussion why the reported result is good for the AUC metric vs the IoU metric. Finally, I appreciate the comparison with compexp in terms of computational time, no reasoning is provided for it, merely the times are reported. Some more discussion as to why is compexp slow? what is the bottleneck that this work gets rid off would help appreciate the results. This is particularly important since the authors use the same algorithm compexp uses to compose concept literals into boolean formulas to form explanations.

4. **missing details in fine-tuning experiment:** In 5.2 what are the Caltech concepts? The imagines concepts were taken from wordnet, is the same Wordnet employed for the Caltech concepts. This should be clearly mentioned in the Appendix for reproducibility. The authors remark "By simply linking the most suitable representations from the latent layer to the output class logit using our approach, we were able to attain a substantial non-random accuracy". Could you please specify what most suitable representations mean here? how were they linked? Was a linear layer trained on top of the most suitable representations? Without these details it is hard to appreciate the merit for this experiment.

5 **more qualitative examples:** The paper makes several claims such as can detect spurious correlation, AUROC better reflects visual similarity than IoU and neuron explanations can be used to handcraft small circuits for classification on target datasets without any training (which I think is a great use-case!). However, only one or two examples are provided for each case. This raises concerns for cherry picking, the authors should provide many more examples in the supplement for readers to further substantiate the claims. Moreover, if possible the authors should also do quantitative experiments to validate their claims, perhaps a mechanical turk study for the statement "AUROC better reflects visual similarity that IoU".

6 **statistical significance of explanations**: The authors claim an advantage of AUROC over IoU is that their metric provides a measure of statistical significance. I think this is an interesting point but I am not an expert on hypothesis testing and the paper does not mention how the AUROC is a measure of statistical significance to appreciate this point. A paragraph dedicated to this would strengthen the paper in my opinion.

**Minor Points**
I think it should be made clear in the main text (as opposed to the supplementary) that for CNN filters, the authors treat the output as a scalar by averaging the feature map as opposed to thresholding the feature map to get highest activating regions (as in prior work). This is an important point and should be bought up before doing the comparison with previous work in the main text.

---

> ### Author Rebuttal · Authors · 2023-08-10
>
> Dear Reviewer UsfE,
> We extend our sincere gratitude for the dedicated time and the insightful comments you have shared with us. Your thorough review has proven invaluable in refining our work, and we deeply appreciate your efforts.
>
> In response to the questions and concerns you raised, we would like to provide the following clarifications:
>
> *Q1: Incorrect Prior Work/Missing References*
>
> **Answer**
> We acknowledge your observation and have made the necessary amendments. The incorrect reference to the MILAN method has been rectified, and we have included the reference to the CLIP-Dissect paper in the updated version of our manuscript.
>
> *Q2: Missing Details in Figures*
>
> **Answer**
> Thank you for pointing out this aspect. We have taken your feedback into account and have augmented the figures with additional clarifications. In Figure 2a, orange signifies activation of data points within the explanation, while blue represents data points that do not belong the explanation.
>
> *Q3: More Rigorous Comparison to Prior Work*
>
> **Answer**
> Your concern is well-received, and we have undertaken additional quantitative experiments, as detailed in our comprehensive global response to all reviewers. We acknowledge the intricacies of comparing our approach to prior methods, especially considering the absence of a standardized baseline. Our rationale for favoring the AUC approach over IoU-based methods is extensively explained in the global response. We have observed that, in certain cases, explanations with 0 IoU scores correspond to classes that systematically maximize neuron activation. For explanations with non-zero IoU, we have demonstrated a significant positive correlation with AUC scores. The AUC approach, offering advantages in terms of speed, dataset independence, and wider applicability, emerges as the more suitable choice, in our opinion.
>
> *Q3.2: Reason for Faster Computation*
>
> **Answer**
> Indeed, the optimization algorithm was inspired by the CompExp method. In comparison, INVERT's efficiency stems from operating on binary vectors (0 for data points outside the explanation, 1 for those within), whereas CompExpl conducts logical operations on high-dimensional binarized masks, incurring greater computational and memory costs.
>
> *Q4: Missing Details in Fine-tuning Experiment*
>
> **Answer**
> We acknowledge the validity of this comment and have made efforts to provide a more comprehensive explanation of this experiment in the updated version of our paper. Notably, the CalTech classes exhibit differences from those found within the ImageNet dataset. In our examination, we pinpointed 46 classes that share identical names across both datasets. Our approach involved a search for the latent layer representation within the ImageNet-pre-trained model that exhibited the highest AUC in relation to the ImageNet counterpart of the CalTech class. To illustrate, if we consider the CalTech “barn” class, we identified, within the latent layer of the model, a neuron with the most substantial AUC towards the ImageNet “barn” class. We then used the output of this representation as a logit-prediction for the CalTech “barn” class. Our findings underscore the efficacy of this relatively straightforward methodology, yielding commendable performance levels on the CalTech dataset.
>
>
> *Q5: More Qualitative Examples*
>
> **Answer**
> Your suggestion resonated with us, and we have not only provided additional quantitative results and explored emerging circuits but also augmented our findings with local explainability methods. Moreover, we have introduced extra examples showcasing "handcrafted" circuits. All of these enrichments are detailed in the attached PDF.
>
> *Q6: Statistical Significance*
>
> **Answer**
> Your observation regarding statistical significance is appreciated. We concur that IoU scores can be misleading, lacking in the ability to distinguish random occurrences from systematic patterns. The AUC measure offered by INVERT not only delivers interpretable measure (AUC) but also accompanies it with a p-value for a statistical test (i.e. testing hypothesis “AUC of given explanation = 0.5”), determining the dissimilarity between distributions of activations for explanations and non-explanations. This discussion is further elaborated in our global response.
>
> *Q7: Minor Points*
> We have taken heed of your feedback on minor points and have promptly rectified them in the updated version of our manuscript.
>
> We would like to express our sincere gratitude for your insightful questions and comments. We are hopeful that the evidence we have provided will positively contribute to the re-evaluation of our work. Your thorough review has been invaluable in shaping the enhancement of our manuscript

---

> > ### Comment · Reviewer_UsfE · 2023-08-13
> > **Thank you for your rebuttal**
> >
> > Thank you for your detailed response. I have read it carefully, along with all the other reviews and the corresponding rebuttals. While I think the improvements suggested in the rebuttal would certainly improve the presentation in the paper and help validate some of the statements made in the paper, I maintain my score since in it's current form I do not believe the paper is ready for publication. The suggested changes by the authors in this rebuttal, would require significant changes to the writing (added details about experiments which is crucial for reproducibility, added details about the statistical significance of their scores along with experimental results to back them up) which will not get reviewed for quality control.

---

### Official Review · Reviewer_6wCX · 2023-07-08

**Soundness:** 3 good
**Presentation:** 3 good
**Contribution:** 3 good
**Rating:** 5
**Confidence:** 2

**Summary:**

This paper addresses the limitations of existing global explanation methods for Deep Neural Networks (DNNs) and proposes a new approach called Inverse Recognition (INVERT). Unlike previous methods, INVERT does not rely on segmentation masks, provides scalable interpretability, and enables statistical significance testing. The paper demonstrates the applicability of INVERT in various scenarios, including interpreting representations affected by spurious correlations and revealing the hierarchical decision structure within DNN models.

**Strengths:**

First of all - I'm not an expert in this field, so take everything with a grain of salt. That said, I like this approach for its simplicity. Also, I particularly like the experiments on detrimental representations (section 5.1) and on "finetuning without training" (section 5.2), which show that the proposed method has practical implications.

**Weaknesses:**

See Questions.

**Questions:**

- What is the reason for selecting very specific neurons? Were they found empirically, is this based on previous work, or is there some other reason?
- How applicable is this method to modern ViT-based architectures, such as those used in CLIP?
- Is it possible to have an "open vocabulary" for the concepts? One major shortcoming I see is that we have to use a fixed set of concepts to explain representations (although we can combine them to some extent using logical operators, as discussed in the paper)

**Limitations:**

not explicitly discussed in the main paper.

---

> ### Author Rebuttal · Authors · 2023-08-10
>
> Dear Reviewer 6wCX,
>
> We extend our sincere appreciation for the dedicated time and thorough evaluation of our manuscript. We deeply value the recognition of the simplicity of our approach and the validation of our experiments' practical applicability.
>
> We will proceed to address the questions that have been raised.
>
> *Question: What is the reason for selecting very specific neurons? Were they found empirically, is this based on previous work, or is there some other reason?*
>
> **Answer:** In our study, we have showcased the versatility of INVERT across various models. While specific neurons such as neuron 154 in Figure 6 were selected due to prior research pointing to its susceptibility in recognizing Chinese watermarks, the neurons chosen for the circuits, both in the original paper and the attached PDF, were not deliberately selected. These examples are meant to illustrate INVERT's performance in authentic scenarios. Our overarching objective is to underline the universal applicability and adaptability of our method across diverse models and neurons.
>
> *Question: How applicable is this method to modern ViT-based architectures, such as those used in CLIP?*
>
> **Answer:** The presented INVERT methodology is universally applicable to any neural representations (neurons) producing scalar outputs, including ViT-based CLIP, or just transformer models. In general our experiments demonstrate successfully the application of INVERT to ViT-based architectures within section 5.2. Additionally, the attached PDF showcases an expanded set of examples rooted in ViT architecture, providing further evidence of our method's application scope.
>
> *Question: Is it possible to have an "open vocabulary" for the concepts?*
>
> **Answer:** So far the dependency on data remains a central constraint for methodologies aiming to explain learned abstractions of neural networks and connecting neurons to comprehensible human concepts. Methods like Network Dissection and Compositional Explanation of Neurons necessitate images with object masks. INVERT mitigates this dependency by requiring image labels only. This not only makes the data collection easier but also accelerates computational processes. As detailed in section 5.1 of our paper, the set of concepts can be expanded by merging datasets from diverse sources to encapsulate a wider array of concepts. To entirely liberate the approach from data dependency, a critical, presently undiscovered step akin to the transition from supervised to unsupervised learning is requisite—a challenge we aim to tackle in forthcoming research.
>
> Furthermore, we have augmented our global response with new qualitative illustrations and comprehensive quantitative evaluation results. We thank Reviewer 6wCX very much for their valuable feedback, which greatly improved the quality of our work and hope that these additions will persuade Reviewer 6wCX to reconsider and possibly improve the evaluation of our work.

---

> > ### Comment · Reviewer_6wCX · 2023-08-19
> >
> > Dear authors,
> >
> > thank you very much for your response, which I appreciate. In general, I would like to see a stronger focus on the analysis of modern transformer-based architectures but I understand the need to compare to prior methods mainly evaluated on ResNets (which are still widely used in practice).
> >
> > I will keep my original rating.

---

### Author Rebuttal · Authors · 2023-08-10

We extend our deepest gratitude for your invaluable dedication and expertise in assessing our work. The thoughtful feedback of all the reviewers has been instrumental in refining our research and elevating its quality. The positive reception of our work has been immensely gratifying, and with this global rebuttal, coupled with the attached PDF, we aim to address your questions and provide a comprehensive overview of our responses. Additionally, we address all the reviewers individually.

**Evaluation**

To address concerns about additional evaluations, we conducted an extensive quantitative assessment. We acknowledge the complexities of comparing our method to existing approaches due to the lack of a standardized baseline for the evaluation of global explanation methods. Our motivation was to demonstrate the limitations of IoU-based explanations. In Figure 1 of the attached PDF, we present a case where explanation yielding 0 IoU score is better aligned with the explanation goal. We provide evidence of IoU-based explanations resulting in low neuron activation, while INVERT achieves notable activation even when IoU scores are 0. Furthermore, we highlight a correlation between IoU and AUC scores in non-zero IoU cases across multiple models and layers (Table 1). This, coupled with INVERT's computational efficiency, lack of dependency on masked datasets, interpretable scoring measure (AUROC), and wider applicability, positions INVERT as a robust alternative to IoU-based methods.

**Additional Qualitative Experiments**

In the attached PDF, we have introduced new qualitative experiments to strengthen our claims and demonstrate the practicality of our proposed method. We enhanced existing figures with GradCam explanations (Figures 2 and 3). Additionally, we introduced a visualization for a new circuit (Figure 4) and included new "handcrafted" circuits for the ViT model (Figure 5), supplementing the example from the original paper (Figure 7).

**Statistical Significance**

IoU-based explanations often suffer from reporting small IoU scores for highest-IoU explanations, raising concerns about random coincidences (for example, Figure 1 in the attached PDF). Notably, there exists no statistical measure to test the hypothesis $H_0:$ IoU$ = 0.$ INVERT's AUC-based method naturally connects to the Mann-Whitney-Wilcoxon test statistic, providing a means to test the hypothesis "AUC of a given explanation = 0.5." Table 2 furnishes a sanity check for INVERT, demonstrating that, with random explanations, INVERT yields AUCs $\approx 0.5$.

After diligently addressing the valuable feedback provided by the reviewers, we have implemented a series of updates to our paper. These revisions encompass new figures and experiments, as well as the rectification of minor errors. With these clarifications and evidence, we trust that we have addressed potential queries regarding our work. We remain optimistic that this comprehensive information will positively influence the re-evaluation of our paper.

---

### Decision · Program_Chairs · 2023-09-21

**Decision:**

Accept (poster)

**Comment:**

The authors present an approach for interpreting neural representations by mapping them onto human-understandable concepts.
After considering the rebuttal and internal discussions, three reviews are marginally positive and one recommends rejection. While there was no particular excitement about this work, there are also no critical conceptual flaws. On the negative side are mostly experimental and presentation issues: exploring more powerful network architectures, broader experimental evaluation, the study so far being limited to only selected neurons, and the discussion of related work. The authors have put tremendous effort into the rebuttal and showed how to address most critical issues already. Therefore, there appears to be a clear path from here to a camera-ready version that addresses all reviewer concerns.
-> accept. Congratulations!
It is imperative that the final version implements all that has been shown and promised in the rebuttal and addresses all reviewer comments.